# Development of a Highly Efficient Shoot Organogenesis System for an Ornamental *Aeschynanthus pulcher* (Blume) G. Don Using Leaves as Explants

**DOI:** 10.3390/plants11192456

**Published:** 2022-09-21

**Authors:** Honglin Yang, Honglin Yuan, Cunmei Du, Liyun Liang, Meiling Chen, Lijuan Zou

**Affiliations:** 1Ecological Security and Protection Key Laboratory of Sichuan Province, Mianyang Normal University, Mianyang 621000, China; 2State Key Laboratory of Crop Gene Exploration and Utilization in Southwest China, Sichuan Agricultural University at Wenjiang, Chengdu 611130, China

**Keywords:** *Aeschynanthus pulcher* (Blume) G. Don, leaf explant, embryogenic callus, propagation, shoot regeneration

## Abstract

*Aeschynanthus pulcher* (Blume) G. Don, the “lipstick plant” is a prized ornamental plant with distinctive flowers. Here, we introduce a novel in vitro regeneration method for *A. pulcher* using leaf explants and an optimized combination of phytohormone plant growth regulators (PGRs). The optimal conditions for shoot regeneration included 1 mg L^−1^ polyvinyl pyrrolidone (PVP) plus 3 mg L^−1^ thidiazuron (TDZ), inducing a response rate of 82.4% and a shoot/explant ratio of 38.6. When the Murashige and Skoog (MS) medium contained indole-3-butyric acid (IBA) alone, leaves first differentiated into adventitious roots and then adventitious shoots. Leaves cultured on MS medium containing 1 g L^−1^ PVP, 3 mg L^−1^ TDZ, 5 mg L^−1^ casein, and 0.1 mg L^−1^ α-naphthaleneacetic acid (NAA) for 30 d exhibited the highest embryogenic callus (EC) induction rate (95.6%). The optimal shoot proliferation coefficient (21.5) was obtained when shoots derived from EC were cultured on the same medium as that used for EC induction for 5 weeks. The most effective medium for rooting of elongated shoots was MS medium containing 1 g L^−1^ PVP, 5 mg L^−1^ casein, 3 mg L^−1^ 6-benzyladenine (BA), and 0.1 mg L^−1^ NAA, and the number of roots reached 18.8. The regenerated plants grown in a greenhouse had 100% survival following one week of hardening. Overall, our effective and efficient propagation method should result in shortened culture periods and reduced production costs, allowing for the future selective breeding and genetic improvement of *A. pulcher*.

## 1. Introduction

*Aeschynanthus*, an ornamentally important genus of subtropical plants, encompasses approximately 160 species within the Gesneriaceae family [1]. Originating in Southeast Asia, *Aeschynanthus* eventually spread into mainland Southeast Asia and Malaysia. *Aeschynanthus* species are prized primarily for their vivid purple, red, orange, or yellow flowers [2]. *Aeschynanthus* species also make popular house plants, thriving in pots and hanging baskets [3]. Additionally, several species of *Aeschynanthus* are known to have traditional medicinal uses [4], including the use of leaves and leaf extracts in the treatment of fall injuries, rheumatoid arthritis, and postpartum depression [5,6].

The “lipstick plant” *A. pulcher* is particularly famous for the bright red, tubular flowers which emerge each spring [7]. This epiphytic species is native to the jungles of Java and has become popular in contemporary China, and across the world [7,8]. This species exhibits highly specialized sexual features and limited capacity for self-fertilization, resulting in low reproductive capacity. Seed propagation is rarely used in production, due to an inherently poor seed setting rate [9]. Because of this, propagation is generally carried out through cutting or layering [10,11,12]. Unfortunately, these surgical propagation methods carry the risk of spreading disease to the propagated plants [13]. 

To address the challenge of producing industry-relevant amounts of high-quality plant material, an in vitro propagation protocol is necessary. There are scarce reports of shoot organogenesis in *Aeschynanthus*. Cui [2] achieved plant regeneration through direct somatic embryogenesis in *A.radicans*. Subsequently, Zhang and Mei established a callus induction and multiplication system of *A. marmoratus* [14], and *A. lobbianus* [15] through medium containing BA and NAA. However, they only found a 6.1 shoot proliferation coefficient in *A. lobbianus*. Overall, efficient in vitro shoot organogenesis is still needed for *Aeschynanthus*. Here, we sought to develop an efficient clonal propagation system for *A. pulcher*, and predicted PGRs will affect organogenesis and EC formation. The results of this work will enable mass commercial production and genetic transformation in *A. pulcher*.

## 2. Results

### 2.1. The Effect of PGR Application on the Induction of Shoot Regeneration from Leaf Explants

We found significant differences between PGR treatments in both the number and quality of regenerated shoots (Table 1). Overall, shoots were successfully regenerated in MS basal medium supplemented with either 3 mg L^−1^ auxins or cytokinins (Figure 1). The thidiazuron (TDZ)-containing MS medium induced the greatest mean number of shoots per leaf explant (38.6), well above the number produced in the 6-benzylaminopurine (BA)-only MS medium (Figure 1E–H, Table 1), although the shoots were compact, dwarfed, and failed to elongate (Figure 1H). Leaf explants exposed to MS medium containing either 3 mg L^−1^ zeatin (ZT), 2,4-dichlorophenoxyacetic acid (2,4-D), or picloram (PIC) became necrotic and failed to form either callus or shoot tissue (Figure 1C,D). In MS medium containing only indole-3-butyric acid (IBA), the differentiation of adventitious roots and shoots was asynchronous. In this medium, roots developed first (Figure 1I) and subsequently turned brown (Figure 1J). A few shoots developed after 74 d (Figure 1K), with a larger amount developing after 96 d (Figure 1L). The overall response rate in the IBA-containing medium was 60.1%, with a rooting rate of 95.4% (Table 1). Based on these results, it appeared that TDZ and BA-containing MS medium were optimal for shoot regeneration, and were used for all successive experiments.

### 2.2. The Synergistic Effect of Cytokinin and Auxin on Organogenesis and Embryogenic Callus Formation from Leaf Explants

Culturing *A. pulcher* leaf explants on MS medium containing either of several PGR combinations resulted in leaf swelling and the subsequent formation of light-green compact callus along the leaf margins (Figure 2A), following by adventitious root formation within 30 d (Figure 2E, red arrow). Morphological examination indicated that the transparent, globular, and smooth-surfaced callus was EC (Figure 2G). Histological examination further confirmed the EC designation, revealing an orderly arrangement of compact cells containing large, prominent nuclei and dense cytoplasm (Figure 2H). However, the rate of EC formation varied depending on the type and concentration of cytokinins present in the medium. The combination of both cytokinins (BA or TDZ) and auxin α-naphthaleneacetic acid (NAA) reliably induced embryogenic callus formation, with the combination of BA or TDZ (3 mg L^−1^) and NAA (0.1 mg L^−1^) exhibiting a response rate of above 95% (Table 2). After 40 d of culturing, the EC present at the edge of the cut leaf surface began to differentiate and regenerate shoot tissue, accompanied by browning (Figure 2B,F,I). After 70 d of culturing, the number of shoots increased (Figure 2C). After 80 d of culturing, dwarf shoots with expanded leaves were well developed (Figure 2D).

### 2.3. Shoot Proliferation, Rooting, and Acclimatization

Because dwarf shoots are not ideal for root formation or transplanting, we separated the clumps of adventitious shoots into smaller pieces and transferred them to fresh medium promoting proliferation and hardening-off. It was found that adventitious roots could be induced on MS medium with NAA combination with TDZ or BA (Figure 3A,B). MS medium containing 0.1 mg L^−1^ NAA and 3 mg L^−1^ TDZ produced the highest shoot proliferation coefficient (Table 2). Higher concentrations of both TDZ and BA resulted in decreased shoot proliferation response and shoot number, with a BA or TDZ concentration of 4 mg L^−1^ resulting in a shoot proliferation coefficient of 10.0 and 11.6, respectively. When culturing for longer than 30 d, these higher concentrations resulted in the formation of many adventitious roots from the shoot bases (Figure 3C–F). Overall, extensive testing indicated that the combination of 3 mg L^−1^ BA + 0.1 mg L^−1^ NAA + 1 g PVP + 5 mg L^−1^ casein was the most optimal induction medium for shoot regeneration of *A. pulcher*, reliably producing healthy plants. From these plants, well developed roots and shoots containing 4–6 leaves were transferred to a soil mixture for further grow-out in the greenhouse (Figure 3G). Following one week of hardening-off, these plants exhibited 100% survival (Figure 3H). Five weeks after transplantation, the acclimatized plants exhibited normal growth and produced new leaves (Figure 3I).

## 3. Discussion

Plants in the genus *Aeschynanthus* are prize ornamentals, but can be difficult to propagate through seed. Because of this, *Aeschynanthus* plants are typically propagated through cuttings, which can lead to disease. Leaf-based regeneration can reduce disease incidence as well as the likelihood of genetic variation [16]. Until now, micropropagation approaches using leaves as explants have been used to successfully regenerate *A. radicans* [2], *A. marmoratus* [14], and *A. lobbianus* [15] plants, but not *A. pulcher*. Here, we successfully regenerated *A. pulcher* plants from leaf explants using a culture medium containing an optimized combination of PGRs, through both direct organogenesis and indirect morphogenesis. The addition of 1 g L^−1^ PVP and 5 mg L^−1^ casein to the MS medium alleviated explant browning and increased the growth rate.

### 3.1. The Effects of Various PGRs on Organogenesis

We found that exposure to either 3 mg L^−1^ ZT, 2,4-D, or PIC failed to induce adventitious shoots. In contrast, the addition of 3 mg L^−1^ BA, TDZ, or IBA positively influenced shoot regeneration on the *A. pulcher* leaf segments. The cytokinins BA and TDZ are commonly employed in the culture of members in the Gesneriaceae family, and they have shown excellent regeneration potential in *A. radicans* [2], *A. lobbianus* [15], and *Sinningia hybrida* [17]. We found that compared with TDZ, BA induced fewer dark green buds. TDZ is derived from phenyl urea and is well established as one of the most effective PGRs [18]. Our results suggested that TDZ may be critical for the efficient regeneration of large numbers of shoots from leaf explants, in agreement with studies in *Cibotium barometz* [19], *Salvia plebeia* [20], *Portulaca pilosa L.* [21], and *Caryopteris terniflora* [22]. On IBA-containing medium, the development and shoots and roots proceeded asynchronously [23]. The auxin IBA is essential for root development in plants, including the formation of adventitious roots, development of lateral roots, and elongation of root hairs [24,25]. IBA is also known to be essential for the formation of apical hooks, the expansion of cotyledons, and shoot development generally [24,26,27]. Based on our results, it appears that IBA can also induce adventitious shoot formation (Figure 1I–L).

### 3.2. The Synergistic Effect of Cytokinins and Auxins on Callus Formation

Previous research has shown that callus formation is induced through the exogenous application of auxins and cytokinins [28]. Indeed, it appears that cytokinin and auxin work synergistically to induce callus formation [29,30]. Furthermore, it appears that at certain concentrations, callus formation may be induced while direct shoot organogenesis is impeded [31]. Similar observations have been made in other members of the Gesneriaceae family, including *A. radicans* [2], *Saintpaulia ionantha* [32], and *Lysionotus serratus* D. Don [33]. Here, we found that the application of the auxin NAA was particularly beneficial to the formation of callus structures. Morphological examination revealed that the application of NAA produced EC specifically, which was characterized by a compact shape, pale green color, and the presence of small, granular structures. In general, EC may proceed down several morphogenic processes: somatic embryogenesis, nodular embryogenic structures (NESs), or shoot organogenesis [34]. The regeneration of shoots from leaf explants generally occurs through the induction of EC, and often requires the presence of both auxins and cytokinins [35]. Here we found that the induced EC differentiated into adventitious shoots, while roots underwent indirect organogenesis after 40 d of culture, even without medium renewal (Figure 2).

### 3.3. The Synergistic Effect of Cytokinins and Auxins on Callus Formation on Shoot Proliferation

As with the induction of callus formation, a combination of cytokinins such as BA or TDZ and auxins such as NAA is often used to induce shoot proliferation in plants, including in plants belonging to the genus *Aeschynanthus* [2,14,36,37]. Here we found that the combination of BA or TDZ with NAA exhibited a synergistic effect on shoot regeneration from leaf explants, with shoots proliferating and roots forming naturally. Furthermore, we found that regenerated plants transplanted to the greenhouse showed 100% survival, and no additional rooting experiments were necessary. Similar results have been reported for *S. hybrida* [17] and *S. plebeia* [22]. Additionally, the ratio of cytokinin and auxin being applied can alter their interaction and impose a significant effect on shoot regeneration and proliferation [38,39,40]. We found that the combination of 3 mg L^−1^ cytokinin and 0.1 mg L^−1^ auxin was the most effective in promoting shoot proliferation. In fact, shoot and root proliferation tended to decrease with increasing cytokinin concentration, likely due to toxic substances produced as a cellular response to overexposure to cytokinins. This phenomenon has been noted in the propagation of other plant species, including *S.*
*hybrida* [17] and *S.plebeia* [22].

## 4. Materials and Methods

### 4.1. Plant Material and Culture Conditions

*Aeschynanthus pulcher* (Blume) G. Don was purchased from Horticulture Co., Ltd. In Chengdu, China, and certified by Professor Minghua Luo of Mianyang Normal University, China. Terminal buds of *A. pulcher* were used as explants for primary culture. The buds were cleaned in detergent using a banister brush, and thoroughly rinsed in tap water for 1 h. Cleaned buds were then surface disinfected for 35 s with ethanol (75%, *v*/*v*) and for 6 min with HgCl_2_ (0.1%, *w*/*v*), and thoroughly rinsed with sterilized distilled water. Obtained shoots were placed in medium containing 2 mg L^−1^ BA + 0.1 mg L^−1^ NAA for further experimentation. The culture medium used in these experiments was MS medium [41] containing agar (0.7%, *w*/*v*) and sucrose (3%, *w*/*v*). Prior to autoclaving (15 min at 121 °C), the pH was adjusted with NaOH (1 mol L^−1^) to between 5.7 and 5.8. Explant incubation was carried out using an 8 h:16 h dark light cycle, with a 30 µmol m^−2^ s^−1^ light intensity, and a temperature of 25 ± 2 °C. All PGRs and media were purchased from Sigma–Aldrich (St. Louis, MI, USA). Each treatment consisted of thirty explants and three biological replicates.

### 4.2. Screening PGRs for Their Shoot Regeneration Potential from Leaf Explants

After 30 d of culture, leaves of induced shoots were collected for use as explants. Each explant was aseptically divided into square 0.25 cm^2^ segments (Figure 1). Each leaf segment was placed into an 8 cm Petri dish with an approximate volume of 50 mL. To test for the potential of each PGR to promote shoot regeneration, the culture medium contained 3 mg L^−1^ of either PIC, 2,4-D, IBA, ZT, TDZ, or BA. The rate of browning, response, rooting (%), and shoot number were calculated after leaves were incubated on MS medium for 96 d.

### 4.3. Screening PGRs for Their Embryogenic Callus Formation Potential from Leaf Explants

Sterilized leaves were cultured on MS medium for embryogenic callus (EC) induction and plant recovery. Each excised explant was incubated in a culture dish containing 50 mL of medium containing 0.1 mg L^−1^ NAA + either 2, 3, or 4 mg L^−1^ BA or TDZ + 1 g L^−1^ PVP + 5 mg L^−1^ casein. Cultures were observed weekly, and callogenesis (%) was calculated after leaves were inoculated on MS for 4 weeks. Samples were fixed in FAA Fixative Soution (90 mL of 70% ethanol, 5 mL of glacial acetic acid, and 5 mL of 40% formaldehyde). An alcohol series was conducted to dehydrate the samples, and subsequently, each sample was embedded in paraffin wax and sliced into 10–12 μm sections with a rotary microtome. A light photomicroscope (Leica, Wetzlar, Germany) was used for histological examination.

### 4.4. Shoot Proliferation, Root Formation, and Acclimatization

Adventitious shoots derived from EC were collected and relocated to 6.5 cm (D) × 8.0 cm (H), 250 mL cylindrical culture bottles. The culture media contained the same type and concentration of phytohormones as described in the previous section. The cultures were observed weekly, and the root number was recorded. The shoot proliferation coefficient was quantified after 5 weeks of culture, and calculated as the ratio of the number of shoots present after incubation to the number of shoots present before incubation. Rooted plantlets with 4–6 leaves were removed from culture and gently washed with water to remove any remaining agar. In total, thirty plantlets were collected and transplanted into pots. Each pot contained 1:1:1 (*v*/*v*/*v*) ratio of vermiculite, perlite, and nutrient soil. Transplants were grown under the same cultural conditions as described above. After 5 weeks, plant survival (%) was quantified.

### 4.5. Statistical Analysis

SPSS 18.0 (IBM, Armonk, NY, USA) was utilized to perform all statistical analyses. Significant differences between treatments were assessed using both least significant difference (LSD) tests and one-way analysis of variance (ANOVA), with a significance threshold of *p* < 0.05.

## 5. Conclusions

Here, we introduce a novel in vitro regeneration method for *Aeschynanthus pulcher* using leaf explants and an optimized combination of phytohormone PGRs. Such an effective and efficient propagation method should result in shortened culture periods and reduced production costs. Specifically, we found that the combination of BA and TDZ was crucial for shoot induction and proliferation in this species. Additionally, NAA was found to be effective at inducing for formation of EC, which could differentiate into adventitious shoots through indirect organogenesis. This protocol established in the present study will facilitate in generating uniform propagules, germplasm conservation, and genetic engineering processes in *A. pulcher*. 

## Figures and Tables

**Figure 1 plants-11-02456-f001:**
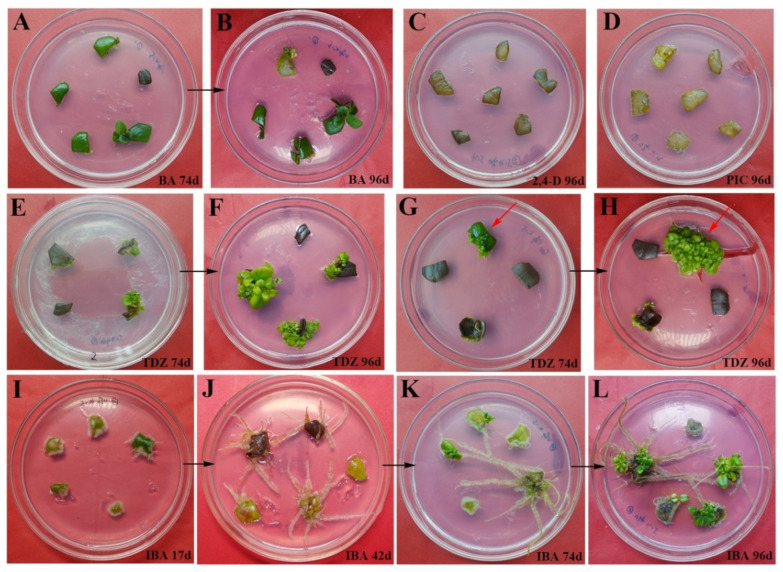
Shoot regeneration from *Aeschynanthus pulcher* (Blume) G. Don leaf explants on MS medium supplemented with different auxin or cytokinin. (**A**) Leaves on BA-containing culture for 74 d; (**B**) Leaves on BA-containing culture for 96 d; (**C**) Leaves on 2,4-D-containing culture for 96 d; (**D**) Leaves on PIC-containing culture for 96 d; (**E**) Leaves on TDZ-containing culture for 74 d; (**F**) Leaves on TDZ-containing culture for 96 d; (**G**) Leaves on TDZ-containing culture for 74 d, leaves blade cut from the middle (red arrow); (**H**) Leaves on TDZ-containing culture for 96 d, leaves blade cut from the middle (red arrow); (**I**) Leaves on IBA-containing culture for 17 d; (**J**) Leaves on IBA-containing culture for 42 d; (**K**) Leaves on IBA-containing culture for 74 d; (**L**) Leaves on IBA-containing culture for 96 d. The concentrations of different PGRs were 3 mg L^−1^ with 1 mg L^−1^ PVP.

**Figure 2 plants-11-02456-f002:**
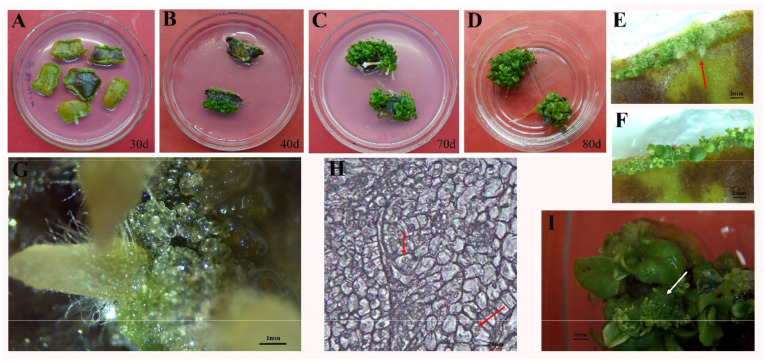
In vitro organogenesis and embryogenic callus induction from *Aeschynanthus pulcher* (Blume) G. Leaf explants on MS medium supplemented with NAA 0.1 mg L^−1^ + BA 2 mg L^−1^ + Casein 5 mg L^−1^ + PVP 1 g. (**A**) Formation of callus and adventitious root following 30 d culture from leaves; Differentiation of adventitious shoot following culture from leaves, 40 d (**B**), 70 d (**C**) and 80 d (**D**), respectively; (**E**) 10× larger view of Figure (**A**), red arrow indicates adventitious root; (**F**) 10× larger view of Figure (**B**), extensive differentiation of adventitious shoots; (**G**) embryogenic callus (EC); (**H**) Histocytological observation of *A. pulcher* EC, the EC with small cells arranged closely and large nuclei (red arrow) (**I**) EC differentiated into adventitious shoots (white arrow). Scale bar = 3 mm (**E**,**F**,**I**), 1 mm (**G**), 20 μm (**H**).

**Figure 3 plants-11-02456-f003:**
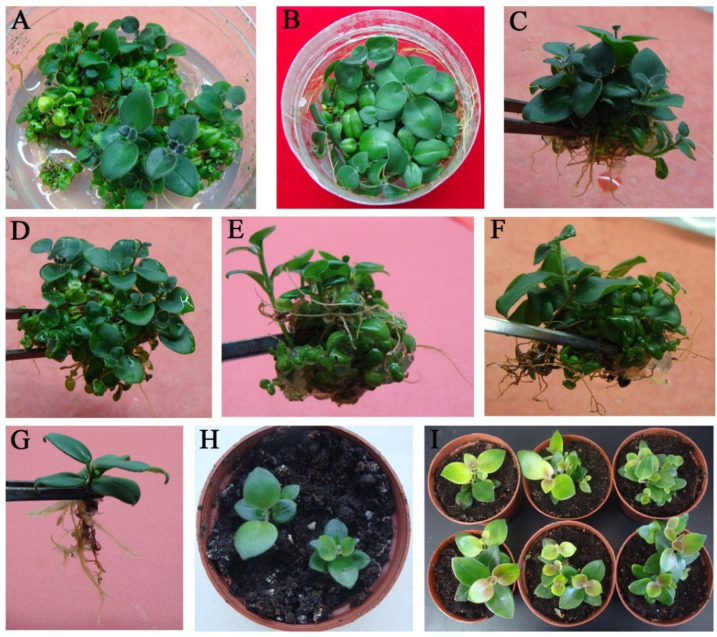
Regenerated shoot proliferation, rooting formation and acclimatization of *Aeschynanthus pulcher* (Blume) G. (**A**) Prolific clustered shoots proliferation on NAA 0.1 mg L^−1^ + TDZ 3 mg L^−1^ + Casein 5 mg L^−1^ + PVP 1 g medium for 5 weeks; (**B**) Prolific clustered shoots proliferation on NAA 0.1 mg L^−1^ + BA 3 mg L^−1^ + Casein 5 mg L^−1^ + PVP 1 g medium for 5 weeks; (**C**,**D**) On NAA 0.1 mg L^−1^ + BA 3 mg L^−1^ + Casein 5 mg L^−1^ + PVP 1 g medium, shoots rooting; (**E**,**F**) On NAA 0.1 mg L^−1^ + TDZ 3 mg L^−1^ + Casein 5 mg L^−1^ + PVP 1 g medium, shoots rooting; (**G**) Rooting condition of single shoots; (**H**) Acclimatized plants from ex vitro rooting after 2 weeks; (**I**) Acclimatized plants from ex vitro rooting after 5 weeks.

**Table 1 plants-11-02456-t001:** Effect of PGRs on shoot regeneration from leaf of *Aeschynanthus pulcher* (Blume) G.

Medium (MS)	Browned (%)	Response (%)	Rooting (%)	Shoots/Explant	Visible Appearance
BA 3 mg L^−1^ + PVP 1 g L^−1^	54.3 ± 1.7 b	50.5 ± 1.9 c	0.0 ± 0.0 b	3.5 ± 0.4 c	No callus, direct induction of shoots regeneration
TDZ 3 mg L^−1^ + PVP 1 g L^−1^	21.5c ± 1.1c	82.4 ± 2.5 a	0.0 ± 0.0 b	38.6 ± 3.5 a	Callus around the leaves, abundant adventitious shoots
ZT 3 mg L^−1^ + PVP 1 g L^−1^	100.0 ± 0.0 a	0.0 ± 0.0 d	0.0 ± 0.0 b	0.0 ± 0.0 d	No callus, no shoots, leaves brown and withered
IBA 3 mg L^−1^ + PVP 1 g L^−1^	40.6 ± 2.0 b	60.1b ± 2.4 b	95.4 ± 3.2 a	14.5 ± 2.4 b	No callus, leaves first differentiate into adventitious root then adventitious shoots
2,4-D 3 mg L^−1^ + PVP 1 g L^−1^	100.0 ± 0.0 a	0.0 ± 0.0 d	0.0 ± 0.0 b	0.0 ± 0.0 d	No callus, no shoots, leaves brown and withered
PIC 3 mg L^−1^ + PVP 1 g L^−1^	0.0 ± 0.0 d	0.0 ± 0.0 d	0.0 ± 0.0 b	0.0 ± 0.0 d	No callus, no shoots, leaves fade to green

Data are presented as means ± standard deviations. Every treatment had 30 leaf explants, the experiments were repeated three times. Different letters within a column indicate significant differences according to least significant difference (LSD) tests (*p* < 0.05).

**Table 2 plants-11-02456-t002:** Effect of PGRs on induced organogenesis from leaf of *Aeschynanthus pulcher* (Blume) G.

Treatment	Callogenesis (%)	Shoot Proliferation Coefficient	Root Number	Observed Results
NAA 0.1 mg L^−1^ + BA 4 mg L^−1^ + Casein 5 mg L^−1^ + PVP 1 g L^−1^	72.6 ± 2.8 c	10.0 ± 1.5 b	12.3 ± 1.0 b	Embryogenic callus; adventitious shoot differentiation; adventitious root formation, dwarf shoot
NAA 0.1 mg L^−1^ + BA 3 mg L^−1^ + Casein 5 mg L^−1^ + PVP 1 g L^−^^1^	95.2 ± 2.4 a	19.4 ± 2.2 a	18.8 ± 0.9 a	Same result as above
NAA 0.1 mg L^−1^ + BA 2 mg L^−1^ + Casein 5 mg L^−1^ + PVP 1 g L^−1^	80.3 ± 4.6 b	13.6 ± 1.8 b	14.2 ± 2.7 a	Same result as above
NAA 0.1 mg L^−1^ + TDZ 4 mg L^−1^ + Casein 5 mg L^−^^1^ + PVP 1 g L^−1^	82.4 ± 3.1 b	11.6 ± 1.3 b	9.2 ± 1.4 b	Same result as above
NAA 0.1 mg L^−1^ + TDZ 3 mg L^−1^ + Casein 5 mg L^−1^ + PVP 1 g L^−^^1^	96.6 ± 1.8 a	21.5 ± 3.7 a	14.4 ± 0.8 a	Same result as above
NAA 0.1 mg L^−1^ + TDZ 2 mg L^−1^ + Casein 5 mg L^−1^ + PVP 1 g L^−1^	84.0 ± 1.7 b	15.6 ± 1.3 b	10.6 ± 0.7 b	Embryogenic callus; adventitious shoot differentiation; adventitious root formation; robust shoot

Data are presented as means ± standard deviations. Every treatment had 30 leaf explants. the experiments were repeated three times. Different letters within a column indicate significant differences according to least significant difference (LSD) tests (*p* < 0.05).

## Data Availability

All data generated or analyzed during this study are included in this published article.

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
