# Peer review of "Development of a Highly Efficient Shoot Organogenesis System for an Ornamental Aeschynanthus pulcher (Blume) G. Don Using Leaves as Explants"

_plants, 2022, doi:10.3390/plants11192456_

Round 1

Reviewer 1 Report

The manuscript is interesting as it contains some useful information on the micropropagation of Aeschynanthus pulcher, ornamental plant originating in Southeast Asia. The manuscript is clearly written and Results and Discussion are equilibrated and pertinent with the outcomes of the investigation.

To be ready for publication in Plants, the manuscript needs some minor revisions. Corrections and suggestions to improve the manuscript are reported directly on the text with related comments that I ask the authors to take into consideration carefully.

In short here, I suggest to the Authors to choose keywords not already present in the title, to improve the indexing of the manuscript. Some examples are given directly on the text.

In Materials and methods, the media composition should be fully reported. For example, paragraph 4.3 fails to indicate the presence of PVP in the medium and in paragraph 4.3 PVP and casein are not specified. Instead, these compounds are incorrectly reported in the tables and in the captions of the figures. This makes understanding tables and figures not easy. As reported directly in the Results, I suggest eliminating the unnecessary from the tables. Furthermore, authors should check the value in Tab 1 for the Browned percentage using PIC.

Author Response

Dear Reviewer 1:

Firstly, I would like to thank you for giving us the opportunity to revise. Thank you very much

for your careful modification.

Question 1: In short here, I suggest to the Authors to choose keywords not already present in the title, to improve the indexing of the manuscript. Some examples are given directly on the text.

Answer 1: Thanks for your suggestions, we selected the keywords according to the core content of the article, and we replaced and added keywords not already present in the title according to your suggestion.

Question 2: In Materials and methods, the media composition should be fully reported. For example, paragraph 4.3 fails to indicate the presence of PVP in the medium and in paragraph 4.3 PVP and casein are not specified. Instead, these compounds are incorrectly reported in the tables and in the captions of the figures. This makes understanding tables and figures not easy. As reported directly in the Results, I suggest eliminating the unnecessary from the tables.

Answer 2: Thank you for your very careful review and suggestions. Since PVP and Casein could alleviate explant browning and increase the growth rate in this experiment, we supplemented the culture medium composition fully in 4.3.

Question 3: Furthermore, authors should check the value in Tab 1 for the Browned percentage using PIC.

Answer 3: Thank you for reminding me about the value in Tab 1 for the Browned percentage. I have carefully checked and found that the leaves in PIC medium were only faded to green without browning phenomenon or other reactions, so the browned percentage is 0%.

Author Response

Dear Reviewer 2:

Firstly, I would like to thank you for giving us the opportunity to revise. Thank you very much

for your careful modification.

Question 1: I would appreciate seeing a full analysis of ANOVA (verification of its assumptions). The description of statistical analyzes of results lacks whether there were significant outliers in the groups of the independent variable in terms of the dependent variable, whether the dependent variable was approximately normally distributed for each group, and whether the homogeneity of variances was preserved? It would be good to provide the results of the appropriate tests (the homogeneity of the variances, the assumption of normality or testing for outliers) checking the above issues.

Answer 1: Thanks for your suggestions. We have tested outliers, normality and homogeneity of variance before the analysis of variance, but the results are not presented in the paper.

Question 2: I would suggest that the Authors partially change the keywords so that the keywords do not repeat with the words used in the title.

Answer 2: Thanks for your suggestions, we selected the keywords according to the core content of the article, and we replaced and added keywords not already present in the title according to your suggestion.

Question 3: Figure 1: Photos appear in Figure 1, showing leaf fragments after 74 days of incubation explants on MS medium. Unfortunately, this information is not included in the "Material and Methods" section. It seems to me that in the "M&M" section, it is worth clarifying that only results after 96 days of incubation were considered. The Presenting of images after 74 days after inoculation introduces ambiguity since it is unclear whether these data were also subjected to statistical analysis.

Answer 3: Thanks for your suggestions. We added the picture of 74 days of incubation explants on MS medium to show the completeness of the experiment and the growth process of the plant, to ensure the authenticity of our data. In the "M&M" 4.2 section, these data were also subjected to statistical analysis after 96 days of incubation.

Question 4: It would be beneficial to include the number of days on the A-D images (Figure 2).

Answer 4: Thanks for your suggestions. We added the number of days on the A-D images to show the completeness of the experiment and the growth process of the plant.

Question 5: line 240: What equipment was used to cut the samples embedded in paraffin?

Answer 5: Thank you for your reminding. We use rotary microtome for cutting the samples embedded in paraffin, and supplemented it in 4.3.

Question 6: Conclusions: It doesn't seem legitimate to me to say that: "favorable genes could also be transferred through genetic transformation" based on the presented results. Such a conclusion is too speculative and would require other studies, which I encourage the Authors to do. Nevertheless, it seems that the Authors should rewrite the latter conclusion.

Answer 6: We have changed the wording.

Question 7: Main note, the manuscript needs linguistic smoothing as some sentences do not fully convey the statement's meaning. However, it seems that improving the language would significantly increase the value of the manuscript.

Answer 7: Thanks for your suggestions. Before submitting the paper, we have found a professional company to polish and modify it.

The remaining question:

lines 53-54: There is no need to include the abbreviation expansion or abbreviation itself

because “PGRs” and “EC” were previously described in the abstract.

line 65: According to the rule that we translate abbreviations when they come in the text for the

first time, I would now explain the meaning of “ZT, 2,4-D, or PIC” here.

line 80: “…cytokinin..” two full stops ending sentence.

line 81: “…2,4D” missing hyphen.

Table 2: “95.2±2.4.a” Please remove the full stop between "4" and "a".

“NAA0.1 mg”. Please keep in mind that there should be a space after the NAA and before the

number value.

I recommend in Table 2 simplifying the entry in the "Observed results" column. The same

observations apply to the first five in vitro culture conditions and could be recorded only once.

Figure 2: “+PVP” Please keep in mind that there should be a space.

“(E) 10x larger view of Figure A, black arrow”: Probably the red arrow?

line 117: Only “embryogenic callus” or “EC”. Since the acronym was already defined, its explanation is not necessary.

Figure 3: Please standardize the nomenclature and put spaces after PGRs and before numerical

values, e.g., such as NAA 0.1 and BA 3.0.

line 167: it should be “explants” not “explains”.

line 186: “avenues”? maybe “processes”

line 193: “6-BA” or just BA? “6-BA” has not been explained.

lines 216, 217, 222, 230, 234: “BA, NAA, PGRs, IBA, TDZ, EC”: Since the acronym was

already defined, its explanation is not necessary.

line 238: “FAA” has not been explained.

line 231: “inoculated”? It appears that the word "incubated" should be used as it relates to the

period the explants were put on MS medium and maintained there for either 74 or 96 days.

Instead, the term "inoculated" describes a one-time procedure in which a medium, often liquid,

is inoculated with something, such as a bacterial strain or an ingredient that has been put into

the liquid media.

line 252: What is the “1a”?

Answer: Thanks for your suggestion, we have been revised.

Reviewer 3 Report

Review for paper:  plants-1912675

 The authors used micropropagation for an ornamental Aeschynanthus pulcher (Blume) G. Don using leaves as explants.. The regeneration was efficient only shoot organogenesis. The authors claimed they provided a novel method.

Could the authors show what the applied regeneration method is innovative/novel? In general, if a methodology is developed for other species of the same genus, the same method with modification can easily be applied to another species.

The documentation for embryogenic callus does not convince.  Why have the authors not checked whether somatic embryogenesis is successful? Obtaining viable somatic embryos and then plantlets, i.e. with conversion (viable somatic embryos have root and shoot apex) would avoid the use of rooting media.

There is also no adequate summary in which the authors clearly present the efficient / optimal regeneration way of plants from the leaves. Maybe the diagram would be useful here.

Author Response

Dear Reviewer 3:

Firstly, I would like to thank you for giving us the opportunity to revise. Thank you very much

for your careful modification.

Question 1: Could the authors show what the applied regeneration method is innovative/novel? In general, if a methodology is developed for other species of the same genus, the same method with modification can easily be applied to another species.

Answer 1: Aeschynanthus, encompasses approximately 160 species. Although there are micropropagation protocols for several Aeschynanthus species, methods of propagation of A.pulcher are lacking. The plant characteristics of different species of the same genus are also different. This study presents an effective method of micropropagation of A.pulcher in vitro cultures, both by direct and indirect organogenesis. And in introduction section, we have been revised according to your suggestion.

Question 2: The documentation for embryogenic callus does not convince.  Why have the authors not checked whether somatic embryogenesis is successful? Obtaining viable somatic embryos and then plantlets, i.e. with conversion (viable somatic embryos have root and shoot apex) would avoid the use of rooting media.

Answer 2: First of all, thank you very much for your very professional question. We did succeed in inducing somatic embryos. It is well known that the root initially formed with only one. Since the root was easily broken when we separated the individual plant in the later stage, we added a step of rooting to ensure the survival rate. Both FIG. 2 and Table 2 show the rooting of somatic embryos.

Question 3: There is also no adequate summary in which the authors clearly present the efficient / optimal regeneration way of plants from the leaves. Maybe the diagram would be useful here.

Answer 3: In the conclusion of the article, we clearly summarized that the combination of BA and TDZ was crucial for shoot induction and proliferation in this species. Additionally, NAA was found to be effective at inducing for formation of EC. The optimal concentration selection is shown in the diagram.

Reviewer 4 Report

These are my main comments on the manuscript (plants-1912675) entitled “Development of a highly efficient shoot organogenesis system for an ornamental Aeschynanthus pulcher (Blume) G. Don using leaves as explants”. The manuscript investigates a novel in vitro regeneration method for A. pulcher using leaf explants and an optimized combination of phytohormone plant growth regulators (PGRs).

Following moderate revisions should be incorporated in the manuscript prior to acceptance.

1. I have concerns about the manuscript sections that I believe need to be addressed in order to improve its clarity.

2. Information about other plant propagation techniques and PGRs function are missing in introduction section.

3. A hypothesis for this work is needed.

4. Other revisions could be checked in PDF attached.

Author Response

Dear Reviewer 4:

Firstly, I would like to thank you for giving us the opportunity to revise. Thank you very much for your careful modification. Please refer to the attachment for specific reply.

Table2  Root number
